# A new Lagrangian in-time particle simulation module (Itpas v1) for atmospheric particle dispersion

Matthias Faust[1], Ralf Wolke[1], Steffen Münch[2], Roger Funk[2], and Kerstin Schepanski[1]

[1]Leibniz Institute for Tropospheric Research (TROPOS), Permoserstr. 15, 04318, Leipzig, Germany
[2]Leibniz Centre for Agricultural Landscape Research (ZALF), Eberswalder Str. 84, 15374, Müncheberg, Germany

**Correspondence:** Matthias Faust (faust@tropos.de)

**Abstract.** Trajectory models are intuitive tools for airflow studies. But in general, they are limited to non-turbulent, i.e. laminar flow conditions. Therefore, trajectory models are not particularly suitable for investigating airflow within the turbulent atmospheric boundary layer. To overcome this, a common approach is handling the turbulent uncertainty as a random deviation from a mean path in order to create a statistic of possible solutions which envelops the mean path. This is well known as Lagrangian particle dispersion model (LPDM). However, the decisive factor is the representation of turbulence in the model, for which widely used models such as *FLEXPART* and *HYSPLIT* use an approximation. A conceivable improvement can be using a turbulence parameterisation approach based on the turbulent kinetic energy (TKE) on high temporal resolution. Here, we elaborated this approach and developed the LPDM *Itpas*, which is online coupled to the German Weather Service's mesoscale weather forecast model *COSMO*. It allows for benefiting from the prognostically calculated TKE as well as from the high-frequent wind information. We exemplary demonstrate the model's applicability for a case study on agricultural particle emission in Eastern Germany. The results obtained are discussed with regard to the model's ability to describe particle transport within a turbulent boundary layer. Ultimately, the simulations performed suggest that the newly introduced method based on prognostic TKE sufficiently represents the particle transport.

## 1 Introduction

We are living in a polluted world. All together, our industry, our mobility, and our agriculture cause particulate matter emissions (Taiwo et al., 2014; Pant and Harrison, 2013; Kjelgaard et al., 2004) that decrease air quality and thus impact on human health in affected regions (Kim et al., 2015). But also natural aerosol emissions like from soil erosion by wind can result in dusty haze plumes or mature dust storms which may locally cause poor air quality (Goudie, 2014). In central Europe, dust emissions can occur in agricultural environments. These are driven on the one hand by wind erosion and on the other hand by activities like tillage and harvest (Goossens, 2004). Wind erosion occurs predominantly in plains and on sandy soil under preferable conditions such as high wind speeds, low vegetation cover and low soil moisture. On average the soil loss by wind erosion in Europe is estimated to $0.53 \, \mathrm{t \, ha^{-1} \, yr^{-1}}$ Borrelli et al. (2016), but extreme wind erosion events can be associated with a soil loss of more than $100 \, \mathrm{t \, ha^{-1}}$ (Funk and Reuter, 2006). Furthermore the increasing drought conditions as a result of the climate crisis (Samaniego et al., 2018; Marx et al., 2018) can be expected to amplify soil loss by wind erosion. Other than wind erosion,

dust emission by tillage is not a function of the wind velocity. Since the particles are lifted mechanically the emission depends mainly on the used tool and the soil moisture (Funk et al., 2008). Goossens (2004) estimated 4-times higher dust emissions caused by tillage operations than by natural wind erosion. Whereas wind erosion is considered in aerosol-atmosphere models since multiple decades (e.g., Joussaume, 1990; Tegen et al., 2002; Huneeus et al., 2011), dust emission driven by agricultural activities is so far underrepresented.

But assuming we know the pollutions' sources, e.g. the tilling tractor, the central remaining questions for air quality studies are: (1) Where is it transported to and, consequently, (2) which regions are affected. Thereby, not only the transport direction but also the altitude and the concentration is of relevance. To shed light on the air travel of a polluted (aerosol loaded) air mass, in general the pathway of an imaginary air parcel is traced through the atmosphere. In other words, its trajectory is determined. This trajectory is then considered as representative for the pathway of the associated aerosol plume.

In order to estimate the air parcel's route through the atmosphere, trajectory models use gridded wind fields as provided by numerical weather prediction (NWP) models for their calculations. But, within the atmospheric boundary layer (ABL), wind respective the air flow is characterised by small- and micro-scale turbulent eddies, and it is challenging to track the motion of individual air parcels due to the frequent change in direction (Stohl, 1998). Since NWP models are usually not able to resolve such small-scale processes, it is impossible to calculate an accurate trajectory in a turbulent boundary layer.

A common approach for trajectory-based dispersion studies within the ABL is the application of *Lagrangian Particle Dispersion Models* (LPDMs). Hereby, the main idea is to handle the deviations due to turbulence as a stochastic deviation from the mean transport path. So, a single trajectory becomes one possible solution of the transport in the chaotic system and an extensive set of trajectories statistically describes the range of possible pathways, which can also be seen as particle dispersion. In this term, particles can be considered either as tracers or simple air parcels, as well as aerosol particles with more complex attributes such as density and size.

Widely used LPDMs like *FLEXPART* (Pisso et al., 2019; Stohl et al., 2005) and *HYSPLIT* (Draxler and Hess, 1998; Stein et al., 2015) are designed to simulate tracer transport on a global scale, but there are also mesoscale applications like *FLEXPART-WRF* (Brioude et al., 2013). To characterise and account for the boundary layer turbulence, these models use a rough estimate of turbulence based on boundary layer depth and atmospheric stability (Hanna, 1982). As meteorological input (e.g. wind, temperature, etc.), *FLEXPART* and *HYSPLIT* use output fields from global forecast or reanalysis models, which are typically available at a time-increment of a few hours (typically 3-hourly or 6-hourly). However, in contrast to the output interval of some few hours, the trajectory calculation should be performed at high temporal resolution (time steps of minutes or less) in order to fulfil the Courant–Friedrichs–Lewy (CFL) condition (Seibert, 1993). Consequently, an interpolation between two consecutive times for which the meteorological input fields are available becomes necessary. This is a crucial issue, especially for small-scale applications at short time scales.

One approach to avoid the temporal interpolation is coupling the trajectory calculation with the NWP-Model directly and thus circumventing the import of (external) meteorological fields to the LPDM. This so-called online coupling (or inline coupling) allows for updating all necessary parameters like wind with every NWP-model time step, typically every few seconds. Compared to the above described offline driven system, this approach results in more accurate trajectories like the study of

Miltenberger et al. (2013) has shown. But there are also disadvantages: Online coupled model systems are only able to calculate forward in time; the calculation of backward trajectories is not possible with this approach. And since a NWP model always has to run just in time with the trajectory computation, additional computational resources are needed.

However, to make an accurate prediction of transport and dispersion processes within the atmospheric boundary layer it is worth to online couple the LPDM to a NWP model. This technique is used in the *WRF* application *WRF-HYSPLIT* (Ngan et al., 2015), where *HYSPLIT* runs just in time with *WRF*. The authors found, that the online coupled *HYSPLIT* version produces better results than the offline version for applications with small temporal and spatial scale (Ngan et al., 2015, 2018). In their study, *WRF-HYSPLIT* uses the online coupling mainly to avoid the temporal interpolation of the winds. But information on the state of the atmosphere may also be useful to improve the estimation of the turbulence in the LPDM, in particular the turbulent kinetic energy (TKE) that is handled as a prognostic variable in many NWP-models.

Here, we integrate the TKE-based approach to the standard trajectory module of Miltenberger et al. (2013), which is included in the state of the science NWP model *COSMO* developed at the German Weather Service (Deutscher Wetterdienst, DWD) (Baldauf et al., 2011). Our approach makes use of the three-dimensional TKE-based turbulence scheme (Doms et al., 2011), which provides the prognostic TKE, and the horizontal and vertical diffusion coefficient of momentum. We further developed a dust emission scheme accounting for mechanical dust emission in agriculture based on measurement data (Münch et al., 2020) and implemented a probabilistic dry deposition scheme (Panitz et al., 1994).

In a nutshell, the new LPDM module *Itpas* (In time particle simulation) now allows for an improved representation of individual trajectories in a turbulent boundary layer. *Itpas* further can handle weightless air parcels as well as aerosol particles like mineral dust. Currently, *Itpas* is used with the *COSMO* version `V5.04e`.

The present manuscript is structured as following: First, we introduce the LPDM *Itpas*, the way it was developed and the physics it uses. Afterwards, we present a showcase for a possible application. Here, we used *Itpas* to simulate the dispersion of a soil particle plume emitted by a tractor.

## 2 Methods

### 2.1 Model description

The starting point of our development was the trajectory module developed by Miltenberger et al. (2013). The software package consists of two files: `data_part` and `src_part`. The first file provides information on parameters, global variables and namelist variables, the file `src_part` contains the LPDM code. There is only a weak link between *Itpas* and *COSMO*. We only added a switch to activate the particle module and call statement in the *COSMO* main routine.

The particle model is called at the end of every *COSMO* time step and reads in the most recent wind and turbulence field arrays. Every particle is handled individually by the LPDM, so, as illustrated in Fig. 1, the overall structure is a loop over every active particle. A typical call performs the following steps: First, the particle transport by the mean wind is investigated. For describing the particle's vertical motion, the settling velocity is taken into account for particles with a mass. Afterwards the turbulent disturbance is calculated. Therefore, it is important to know if the particle resides inside the ABL or above.

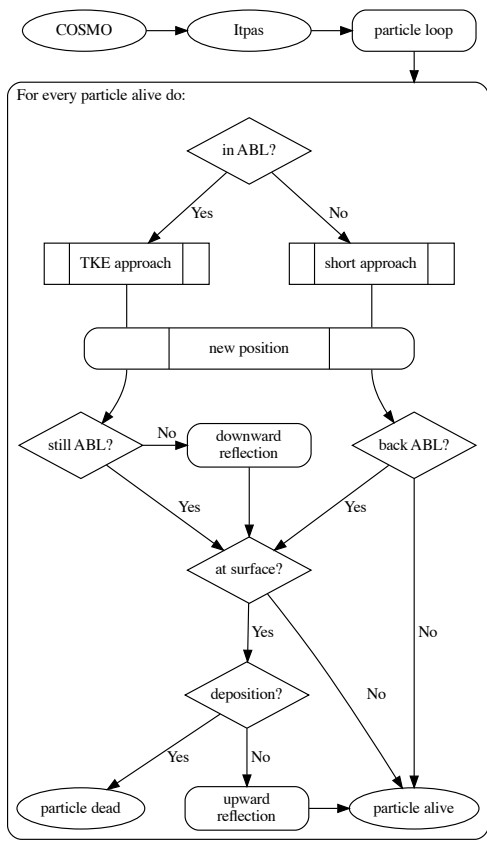

**Figure 1.** Flowchart of the transport algorithm.

- Inside the ABL, the turbulent part of the wind velocity is calculated with a general approach which is presented in detail in the next section. The turbulent wind velocity is integrated with an Euler forward step. This leads to the turbulent part of the particle motion which is added to the mean particle motion. Since the particle is not allowed to leave the boundary layer directly, the model checks if the particle is still inside the boundary layer after the motion. If not, the particle is reflected back into the ABL. If the particle hits the surface, it is decided whether the particle is removed by dry deposition or whether it is reflected back into the atmosphere, based on a probability function (cf. Eq. 17).

- If the particle is above the ABL at the beginning of the calculation, a reduced approach is used to investigate the turbulent part of the particle motion. Afterwards it is checked whether the particle hits the surface, just like described above.

At the end of the loop, the particle is marked either as "dead" or "alive". The particle can "die" because of dry deposition and consequently the removal from the atmosphere, but also by leaving the model domain. A "dead" particle is frozen at its last

position until the end of the simulation. All particles that stay "alive" are ready to run through the particle-loop at the next time step again.

*Itpas* is controlled by a namelist and a start file. In the namelist, important parameters can be defined, like the maximum number of particles, paths for input and output, the seed for the random number generator and the output increment. The start file defines the particle start positions and attributes like size and density. It is possible to define continuously emitting sources and start points that release multiple (thousands) particles. More details on the model setup can be found in the ReadMe-file which is included in the supplement.

The output is written in NetCDF format (Rew et al., 1989) and contains the trajectory positions for every output step as well as the attributes size and density. One of the biggest advantages of the LPDM compared to the Eulerian system is the ability to produce strong gradients and sharp edges of a particle plume. Therefore it was deliberately avoided to transform the trajectories into gridded data for the output. But if needed, it is easy to produce this data form the trajectory information during the post-processing.

## 2.2 Model physics

The Reynolds average of the wind velocity $\boldsymbol{u} = \overline{\boldsymbol{u}} + \boldsymbol{u}'$ describes the true wind velocity $\boldsymbol{u}$ as the superposition of the mean wind velocity $\overline{\boldsymbol{u}}$ with a turbulent disturbing $\boldsymbol{u}'$. Also in the turbulent boundary layer, the major part of the transport is due to advection as a function of the mean wind. For the advective transport, we use the iterative integration scheme (Petterssen, 1956) from the online trajectory module of Miltenberger et al. (2013).

To determine the turbulent proportion of the wind velocity we use a generalised form of Langevin's equation (Lemons and Langevin, 2002).

$$\boldsymbol{u}' = \boldsymbol{a}(\boldsymbol{x}, \boldsymbol{u}, t)\, dt + \boldsymbol{b}(\boldsymbol{x}, \boldsymbol{u}, t)\, \boldsymbol{\xi} \tag{1}$$

The first term of this equation describes the conservation of momentum. The coefficient $\boldsymbol{a}$ is a memory function that depends on the particles position ($\boldsymbol{x}$) and velocity ($\boldsymbol{u}$) at a specific time ($t$). The second term gives a random displacement. The coefficient $\boldsymbol{b}$ contains the potential strength of the turbulence, $\boldsymbol{\xi}$ is a normal distributed random number with a mean value of zero and a standard deviation of one. In that way, every particle gets a random push with the mean strength of the turbulence. The general solution for this selection problem was established by Hall (1975). Looking at $\boldsymbol{u}'$ for the discrete time step $t+1$, Langevin's equation is set up as:

$$\boldsymbol{u}'_{t+1} = \boldsymbol{a}\, \boldsymbol{u}'_t + \boldsymbol{b}\boldsymbol{\xi} \; . \tag{2}$$

The parameter $a$ is defined as the correlation function $\boldsymbol{R}$ (Hall, 1975)

$$\boldsymbol{a} = \boldsymbol{R} = \exp\left(-\frac{dt}{\boldsymbol{\tau}_L}\right) \; . \tag{3}$$

It describes the connection of the model time step $dt$ and $\boldsymbol{\tau}_L$, the characteristic time scale for the Lagrangian dispersion process. The correlation function expresses: If $dt >> \tau_{L;u,v,w}$ then there is no further dependency between the velocities $\boldsymbol{u}'_{t+1}$ and $\boldsymbol{u}'_t$.

The parameter $b$ is defined as:

$$b = \sigma \sqrt{1 - R^2} \; , \tag{4}$$

with $\sigma$ the standard deviation of the wind (Hall, 1975).

Now parametrisations for $\sigma$ and $\tau_L$ are necessary to solve the equation of motion. A common approach uses the TKE ($\bar{e}$), which describes how much kinetic energy is available to generate atmospheric turbulence. Kinetic energy is well known as $\frac{E_{kin}}{m} = \frac{1}{2}v^2$. Based on this the kinetic energy of the turbulent wind fraction is defined as:

$$\bar{e} = \frac{1}{2}\left(\overline{u'^2} + \overline{v'^2} + \overline{w'^2}\right) \; , \tag{5}$$

with the turbulent wind $u'$, $v'$ and $w'$ of the spacial directions. The variance of a variable is defined as the average of the squared fluctuation $\sigma_u^2 = \overline{u'^2}$ (Stull, 2012). This allows parametrising the standard deviation of the wind velocity with the TKE,

$$\sigma = \sqrt{2m\bar{e}} \; . \tag{6}$$

The $m_i$ factors ($m_i \in \mathbb{R}$ with $0 \geq m \geq 1$ and $\sum m_i = 1$; $i = \{u, v, w\}$) describe the weighting of the TKE in each spatial direction, which has a strong dependence on the ambient wind velocity and the atmospheric stratification. In our approach we define the $m_i$ values as the fractions of the mean wind on every model time step.

$$m_i = \frac{|i|}{|\overline{u}|} \tag{7}$$

For the parametrisation of the characteristic time scale, the diffusion coefficient of momentum ($K_m$) is used,

$$K_{m,x} = \frac{1}{2}\sigma_x^2 \frac{\overline{u}}{x} \; . \tag{8}$$

Here, $\sigma_x^2$ is the variance of a spacial dispersed tracer. Taylor (1922) showed that $\sigma_x^2$ can be expressed as:

$$\sigma_x^2 = 2\sigma_u^2 \tau_{L,u} t \; , \tag{9}$$

under the condition that time $t \gg \tau_{L,u}$. The Combination of (8) and (9) with $t = \frac{x}{\overline{u}}$ leads directly to the parametrisation of the characteristic time scale

$$\tau_L = \frac{K_m}{\sigma^2} \; . \tag{10}$$

Now, all parts of the turbulent particle motion are parametrised with available variables of the NWP model. This allows for combining (2), (3) and (4) to the general solution (11), which can be solved with the parametrisations (6) and (10):

$$\boldsymbol{u}'(t) = \boldsymbol{R}\,\boldsymbol{u}'(t - \Delta t) + \boldsymbol{\sigma}\,\sqrt{1 - \boldsymbol{R}^2}\,\boldsymbol{\xi} + (1 - R_w)\,\tau_{L,w}\left(\frac{\partial \sigma_w^2}{\partial z} + \frac{\sigma_w^2}{\rho}\frac{\partial \rho}{\partial z}\right)\begin{pmatrix} 0 \\ 0 \\ 1 \end{pmatrix} \; . \tag{11}$$

For the vertical turbulence, there is a correction necessary that takes into account the vertical variability of the density (Stohl and Thomson, 1999) and the profile of the standard deviation (Legg and Raupach, 1982).

At the top of the boundary layer, there is a hard transition of the wind regime. Inside the boundary layer, there is a turbulent wind regime with possibly strong vertical velocities. Above the boundary layer, the turbulent part of the particle motion is strongly reduced. Therefore a simplified approach can be used for which the standard deviation of the turbulent wind is described by the diffusion coefficient per time step $\sigma = \sqrt{\frac{K_m}{dt}}$ (Stohl et al., 2005), which leads to a turbulent wind velocity of:

$$\boldsymbol{u'} = \sqrt{2\frac{\boldsymbol{K_m}}{dt}} \boldsymbol{\xi} \quad . \tag{12}$$

The model is designed to describe the airborne transport of aerosol particles like mineral dust. As these particles settle by gravity, we need to introduce the gravitational settling velocity ($v_g$), which depends on the particle size and therefore on the Reynolds number ($RE$). For small Raynolds numbers, $v_g$ can be calculated directly (Hinds, 1999):

$$v_g = \frac{\rho_p d_p^2 g C_c}{18\eta} \text{ for } Re \leq 0.4 \tag{13}$$

with the density and diameter of the particle $\rho_p$ and $d_p$, the gravity $g$, the dynamic viscosity of air $\eta$ and slip correction factor (Cunningham correction) $C_c$. For particle settling under condition of high Raynolds numbers ($Re > 0.4$), an empirical approach is used (Hinds, 1999):

$$v_g = \left(\frac{\eta}{\rho_{air} d_p}\right) \exp\left(-3.07 + 0.9935 J - 0.0178 J^2\right) \quad , \tag{14}$$

where

$$J = \ln\left(\frac{4\rho_p \rho_{air} d_p^3 g}{3\eta^2}\right) \quad . \tag{15}$$

Particles near the surface may be removed by dry deposition. Thereby the dry deposition velocity $v_d$ is important, which describes the velocity of the diffusion near the surface. Following Zhang et al. (2001) it is defined as:

$$v_d = v_g + \frac{1}{R_a + R_s} \tag{16}$$

with the atmospheric resistant $R_a$ and the surface resistant $R_s$. Since motion is already calculated individually for each particle, it is useful to handle the deposition individually as well. Therefore we use the deposition probability $W_d$. It describes how likely it is, that a particle remains at the ground once it has hit the surface. An approach for this can be found in Panitz et al. (1994):

$$W_d = \frac{\sqrt{2\pi}\frac{v_d}{\sigma_0}}{F_g + \sqrt{\frac{\pi}{2}}\frac{v_d}{\sigma_0}} \quad . \tag{17}$$

Here $\sigma_0 = 2.5\, u_\star \sqrt{m_3}$ is the standard deviation of the vertical wind right at the surface, where $u_\star$ is the friction velocity. $F_g$ Describes the gravitational part of the dry deposition with $\gamma = v_g \left(\sqrt{2}\sigma_0\right)^{-1}$ and $\mathrm{erf}(\gamma)$ the error function of $\gamma$,

$$F_g = \sqrt{\frac{\pi}{2}}\frac{v_g}{\sigma_0} + \frac{e^{-\gamma^2}}{1 + \mathrm{erf}(\gamma)} \quad . \tag{18}$$

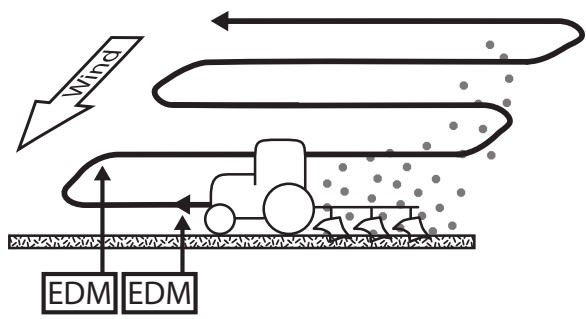

**Figure 2.** Sketch of the experimental setup: the tractor drives in parallel orientated tracks over the field. Downwind the emitted particle plume is measured by two vertically stacked Environmental Dust Monitors.

## 3 Model application

The above described LPDM was developed in order to trace mineral dust particles emitted from arable land during tillage. Other than wind erosion of mineral dust, which is described as a function of the near-surface wind speed, soil particles are entrained mechanically into the atmosphere during agricultural activities on the field like tillage, traffic, fertilisation and harvest. These so-called fugitive emissions may occur independent of the ambient wind speed, but, nevertheless, the ABL dynamics determine the particles' life-time in the atmosphere. Emission potentials for farming activities have been determined in numerous studies (Funk et al., 2008; Holmén et al., 2001; Kjelgaard et al., 2004), but the further fate of the emitted quantities is still relatively unclear. Here, we use *Itpas* to examine the role of the diurnal ABL development to the dispersion of soil dust and organic manure dust particles, which were emitted mechanically during typical field work: The application of dry fertiliser with a spreader and the subsequent incorporation of the fertiliser with a cultivator. The field work was carried out in the framework of the SOARiAL (Spread of antibiotics resistance in an agrarian landscape) project (Thiel et al., 2020), during which airborne particle concentrations were measured as described by Münch et al. (2020).

### 3.1 Field experiment

The experiment took place at an agricultural field near Müncheberg, Germany approximately 50 km east of Berlin, on 31 May 2017 (Münch et al., 2020; Thiel et al., 2020). During this day, the weather over east Germany and west Poland was dry with some shallow cumulus clouds. During the experiment a gentle breeze (~5 ms$^{-1}$) blew from northwest.

The experiment was carried out as following: A tractor drove in parallel lines across the field, orientated perpendicular to the ambient wind (Fig. 2). Downwind, aside the field, the particle number concentration was measured with two vertically stacked Environmental Dust Monitors (EDM 164, GRIMM-Aerosol Technique) at two different inlet heights (1.5 m, 3.8 m). The EDMs measured particle number concentrations at 31 size bins ranging from 0.25 μm to 32 μm.

The experiment was divided into two parts:

**EXP1** Dried chicken manure was spread on the field as fertiliser from 08:50 to 09:45 UTC

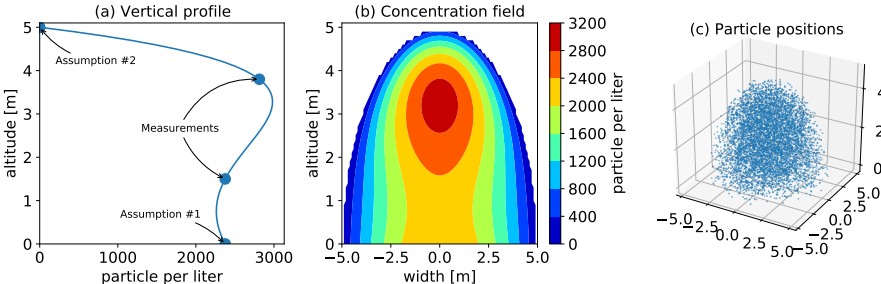

**Figure 3.** Example construction of an idealised particle cloud. a) Vertical profile of the cloud centre, derived from EDM measurements. b) particle concentration in an ideal shaped cloud. c) Particle start positions for *Itpas*

**EXP2** The fertiliser was incorporated into the soil by a cultivator from 11:40 to 12:30 UTC

During both experiments, a particle plume raised behind the tractor; although the tilling entrained far more particles into the atmospheric boundary layer than the fertiliser spreading. Whereas the measurements provide information on the particle number and size distribution and thus characterise the particle plume developing, it remains the question for how long these
particles will remain in the atmosphere and how they will be dispersed. As it can be assumed that the turbulent nature of the convective daytime boundary layer is relevant to the accurate simulation of the dispersion, these two experiments were simulated in order to showcase the strength of our online coupled LPDM *Itpas* driven by prognostic TKE.

### 3.2 Source function

In the framework of this study, dust entrainment was mechanically driven by the tools pulled behind the tractor and not by
wind. A source function was obtained based on the measured particle concentrations in order to describe the dust particle uplift. Therefore we created an idealised version of the particle plume for which we use the EDM data to define a vertical profile of particle concentration. The profile was then multiplied by a half-sphere in order to get a 3D concentration field with which we can define the particle start points for the *Itpas* simulation. Thereby, the particle concentration determines the number of the trajectories starting at the respective point. The particle size is used to assign the corresponding particle properties such
as the diameter, which is relevant for determining the particles residence time in the atmosphere.

In particular, we defined a vertical profile that represents the centre of the tractors particle plume (Fig. 3a). Therefore we used the observation data of the moment when the tractor passed by the measurement device as close as possible. Beside the values of the two measurement heights (inlet heights) two additional assumptions are necessary. First, the particle concentration at ground level corresponds to that at 1.5 m. In general, it seems to be more realistic that the particle concentration is higher at
ground level than at 1.5 m, especially for coarse particles. But since there is no information about the near-surface concentration available, this conservative assumption is the best choice. Second, the particle concentration becomes zero at a height of 5 m. The plume height also depends on multiple properties such as atmospheric conditions, soil texture, soil moisture and the kind of tool pulled by the tractor. But, based on observations and photo documentation of the experiment, 5 m seems to be a good

estimate, at least for this specific case. With these four initial values (two measured & two assumed), the vertical particle number concentration profile of the plume centre was created with a polynomial fit (Fig. 3a).

Now, a normalised spherical concentration field with a radius of 5 m was multiplied with the vertical profile, resulting into a three-dimensional field of the particle concentration (Fig. 3b). From this, the total number of particles inside the plume is determined. For the measurements in the case study that were some billion particles. This number of particles cannot be handled individually within the model simulation and thus needed to be down-scaled. Here we chose a scaling factor of $2\times10^{-7}$. So, one particle in the model represents five million measured particles. In total, we used ~100k particles in EXP1 and ~270k particles in EXP2. All particles are assigned to random positions in the idealised cloud under the condition that particle distribution matches the concentration field (Fig. 3c). This analysis was performed for each bin provided by the EDM data, so the particle size distribution in the model corresponds to the measurements. An extended version of Fig 3 with profiles and concentration fields for the individual bins can be found in the supplement (S1 - S4). With this method we created a bell-shaped cloud of individual particles that we used as model input. Please note that the exact shape of the particle source becomes less important with increasing transport distance.

### 3.3 Simulations

For EXP1 and EXP2 individual simulations were performed. As input data for the NWP-model, we used reanalysis data from *COSMO*, provided by the German weather service. The model domain covers an area from eastern Germany to central Poland including the cities of Berlin and Poznań (lower left corner: 51°N 12°E, upper right corner: 54.4°N 19.3°E) and has a size of $167 \times 150$ grid cells with a spatial grid spacing of 2.8 km and 50 vertical levels.

In the first simulation EXP1 (the spreading of the dry fertiliser), the particles were emitted during the 55 minutes time span from 08:50 to 09:45 UTC at the position 52.477°N, 14.177°E. In EXP2 (incorporation of the fertiliser with a cultivator), the particles were emitted at the same position for 50 minutes from 11:40 to 12:30 UTC. Local time is UTC+2. For both experiments, a continuous release of particles was assumed during the time of emission.

Analysing the simulation output, we focused in particular on the horizontal dispersion, the deposition and the vertical mixing as these three aspects may provide best insights into the model's capability to capture the dispersion of particles within a turbulent boundary layer. These parameters are presented in Fig. 4 and Fig. 5, each with the uppermost panel showing the trajectories indicating the dispersion from a bird eye's perspective. Each line represents the path of one simulated particle trajectory. The second panel shows the same map, but with points indicating where particles were deposited. The last panel shows the altitude of the particles as a function of time, so it visualises the vertical dispersion and mixing of the particles. The colour indicates the particle size ranging from sub-micron particles in light blue to the coarse particles in orange. Please note that smaller particles are overlaid by the larger ones.

In the first experiment, the range of the particle transport was limited to the local surrounding (Fig. 4a). The vast majority of particles have not left this area so that the longest trajectories have a length of about 10 km. The particles spread only a few hundreds of meters in the direction perpendicular to the mean wind . The deposition map (Fig. 4b) shows a colour gradient from west to east, which indicates the coarse particles depositing close to the source while smaller particles are able to travel

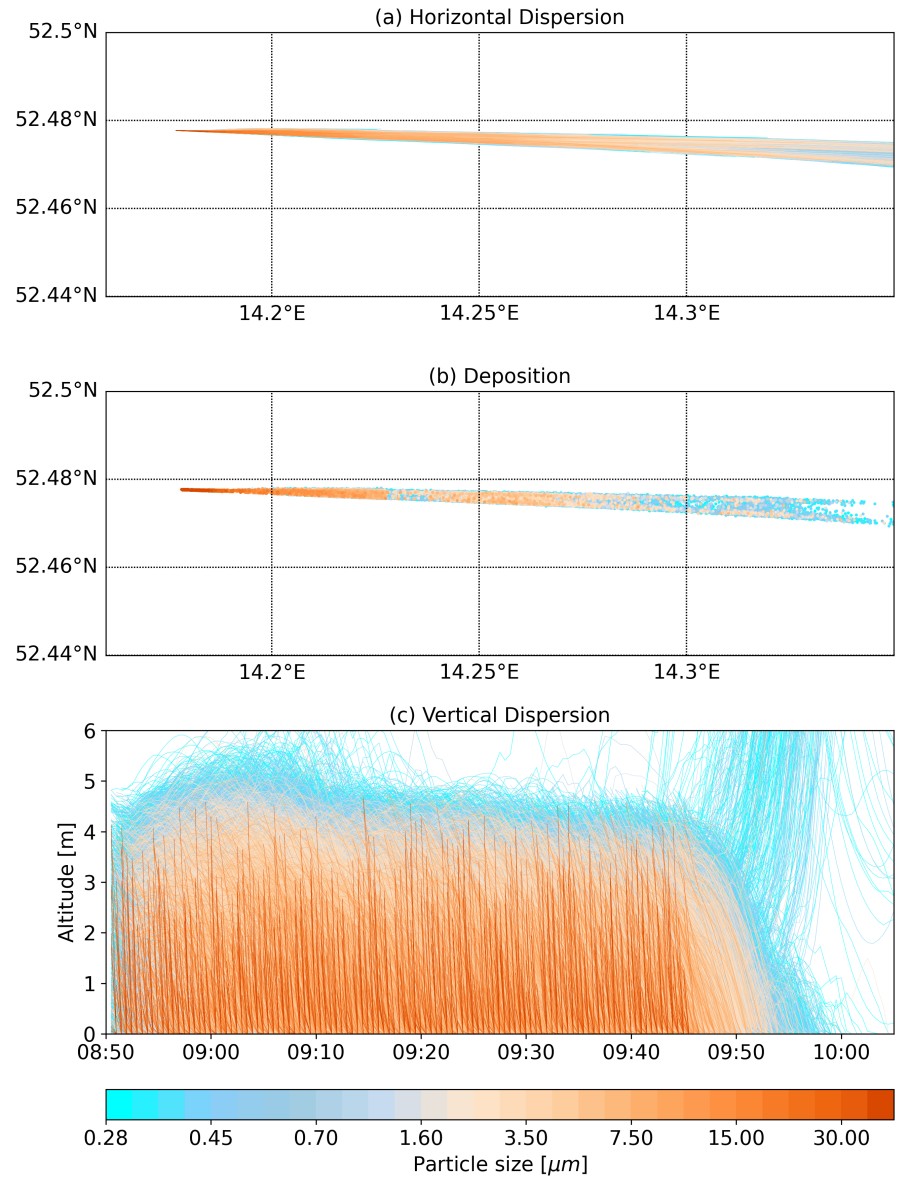

**Figure 4.** Particle motion of EXP1; a) Horizontal trajectories; b) Deposition points, c) Vertical trajectories.

longer distances. This becomes even clearer from time series of the vertical dispersion (Fig. 4c). Please note that the altitude axis is restricted to 6 m. The particles were not lifted into higher levels and fell down more or less quickly after entrainment depending on their size. At the upper edge of the plume, it is visible that the smallest particles shown in light blue form a kind of wave-like structure, which is linked to the random fluctuation that every particle experiences. Because of gravity forces, the amplitude of these waves becomes smaller with increasing particle size and disappears totally for the heavy particles that fall

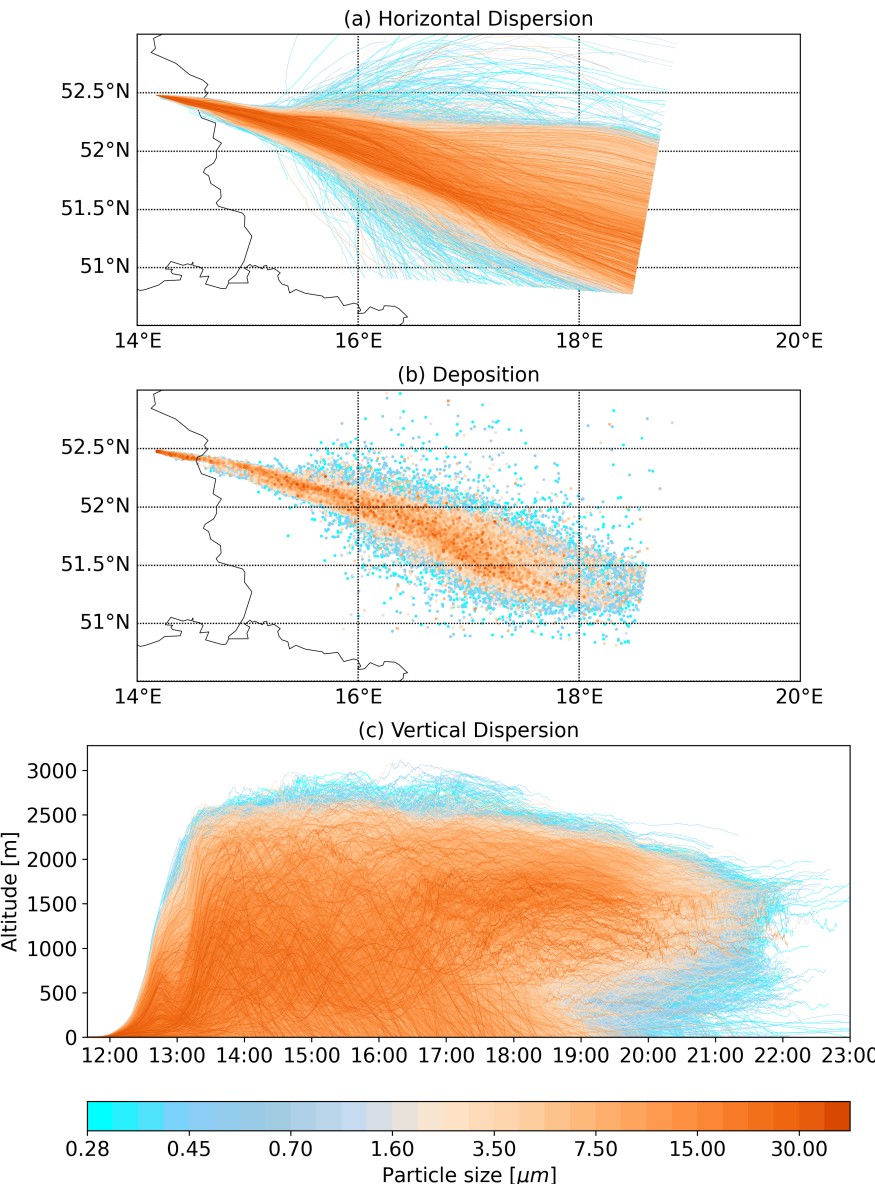

**Figure 5.** Particle motion of EXP2; a) Horizontal trajectories; b) Deposition points, c) Vertical trajectories.

down directly. At the end of the displayed time range, individual particles experience an upward movement and travel further on. These particles perform a similar movement like the particles in EXP2.

Only a brief look at Fig. 5 is necessary to notice that the situation changes drastically in EXP2. Beginning again with the horizontal dispersion (Fig. 5a) it is evident that the particles are going into long-range transport mode. The particles drift

5     evenly apart as they cross the German-Polish border. Above west Poland (~15.5°E), the spread of the trajectories widens and

thus the area covered by the particle plume (sum of the trajectories) increases significantly. Particularly affected hereof are the small particles (shown in light blue colour shades) which partly deviate strongly from the centre of the plume. The trajectories terminate above central Poland where the edge of the simulation domain is located. The most obvious feature on the deposition map (Fig. 5b) is the tendency of the small particles to spread further away from the core plume than the large particles do.

But more interesting are the locations where the large particles deposit. It is apparent that these particles tend to fall down early during their journey. An indication for this is the dark orange stripe from the source to the German-Polish border (Fig. 5b). But also further on over Poland, a remarkable amount of dark orange dots can be found on the deposition map. This is an indication of an efficient vertical transport that was able to uplift the coarse-mode particle fraction. The particle uplift (Fig. 5c) can be described by three phases: the first phase is characterised by the emission of the dust particles (11:40 - 12:30 UTC),

during which the particles slowly rise up to a couple of hundreds of meters above ground level. Then, at around 12:20 UTC, the particles experienced a strong uplift onto a level of 1200 m (second phase), visible as shoulder in the uplifting trajectory pattern (c.f. Fig. 5c). Finally, the particles experience a second strong lifting up to 2500 m above ground at around 13:00 UTC (third phase). The upper edge of the plume can clearly be identified as the top of the boundary layer. During the afternoon, the particles were mixed over the whole depth of the ABL. At 18:00 UTC (Local time: UTC + 2) again a transition in the

distribution pattern is visible. Below 800 m altitude the particles slowly sink to the ground while the ones above this height start to travel on constant levels with strongly reduced vertical mixing. This process forms the overhanging lip of the vertical pattern in the last third of the plot.

## 4   Discussion

### 4.1   Model results

To completely understand the behaviour of *Itpas*, it is necessary to have a look into the atmosphere that was produced by *COSMO*, as illustrated in Fig. 6. It shows a temporal cross-section through the atmosphere or more precisely, the air column at the mean particle position. During the morning, this is the start point of the plume (the short-range transport from EXP1 is ignored). During the afternoon, the displayed air column follows a mean trajectory from the plume of EXP2. The mean trajectory travels in the centre of the particle plume and reaches the eastern edge of the model domain around 21:30 UTC. For

the remaining time it rests at its terminal position. The altitude of the mean trajectory is marked in the plot as red line together with the standard deviation in pale red. The main variable displayed in the cross-section is the virtual potential temperature $\theta_v$ in colour shades from deep blue to dark orange. $\theta_v$ is a combined measure of temperature and moisture, weighted with the level height. Therefore it is well suitable to classify the atmospheric stratification. Vertically increasing values of $\theta_v$ indicate stable, decreasing values unstable stratification. Fig. A1 in the Appendix section shows the $\theta_v$ profile for the consecutive hours

of the day and may help to interpret Fig. 6.

A closer look into the near surface atmospheric conditions helps to understand the model behaviour in the morning hours (EXP1), during which the particles were not able to lift. A detailed view on the bottom-most 50 m can be found in the upper left corner of Fig. 6. Here, it is visible that $\theta_v$ increases at the levels above the surface, which forms a small stripe of stable

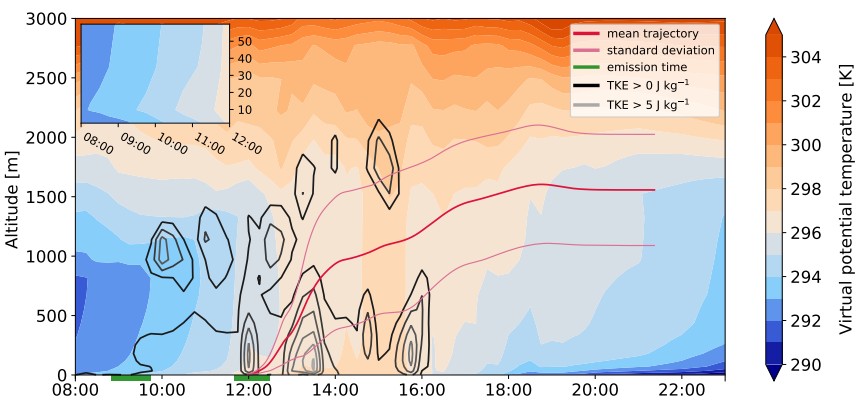

**Figure 6.** Atmospheric conditions for the particle dispersion. Cross-section of the virtual potential temperature and TKE along the mean particle position of both experiments.

stratification. In a stable atmosphere, the vertical motion is inhibited, so no momentum is available to uplift the particles. During the morning hours, the vertical $\theta_v$ gradient becomes weaker and disappears until noon. But not only the atmospheric stratification is important for the vertical mixing, the overall driver of turbulent momentum is still the TKE. Now, focusing on the main plot of Fig. 6, the available TKE is drawn in dark grey isolines. The hours of emission during the different experiments

are marked in green on the time axis. Besides the limited vertical motion, there is no TKE available at the beginning of EXP1, although a slight increase of the TKE during the end of EXP1's emission time occurs, which may provide enough momentum to push single particles upwards out of the stable surface layer (cf. Fig. 4c). However, even if the vertical particle motion in EXP1 behaves reasonable, the horizontal movement of the particles should be interpreted carefully. In the simulation, within a radius of up to 10 km around the source, the particles are uplifted less than 5 meters above the surface. This close to the ground

level, the COSMO-model can not provide reliable information of the horizontal wind do to its vertical resolution. Furthermore, the spatial grid resolution of 2.8 km is relatively coarse compared to the travel range (10 km). Additionally, small surface structures like buildings, trees or forests are not included in the simulation. Thus, EXP1 should be considered as a case without particle transport. In essence, the dust particles emitted by the tractor are whirled up but then deposit soon after emission.

Later during the day for EXP2, we can see changes in both the surface-near TKE and the $\theta_v$ profile. At noon, a decrease of

$\theta_v$ up to a height of at least 500 m is notable, which indicates unstable conditions in this layer. This allows the particles to rise, whereby the TKE induces the vertical mixing. During the afternoon, the unstable layer extends up to 2000 m. Above 2500 m a strongly increasing gradient of $\theta_v$ indicates the top of the ABL. This gradual development of the ABL can also be observed from the trajectories (cf. Fig. 5c between 12:00 and 13:00 UTC).

Between 13:00 UTC and 14:00 UTC, the lower half of the particles (below the mean trajectory), passes a region with strongly

increased TKE and the particles feel momentum which enables them to shear out strongly from the main plume. This situation corresponds spatially and temporally to the enhanced spreading of the trajectory plume above west Poland (cf. Fig. 5a).

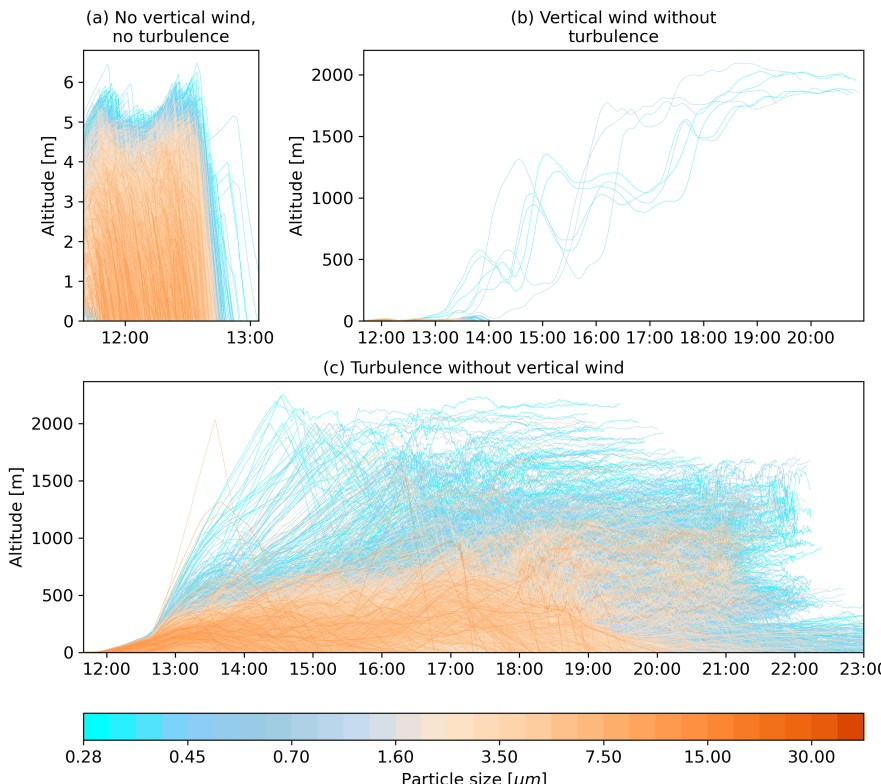

**Figure 7.** Vertical dispersion in a artificial setup: a) without vertical wind nor turbulence; b) vertical wind only; c) turbulence only

After 16:00 UTC the temperature at the surface is decreasing and the ABL collapses forming the transition into a nocturnal boundary layer structure: A stable surface layer forms which deepens during the evening into the typical stable nocturnal boundary layer. The particle plume gets divided into two parts, one remaining within the stable ABL and one above. Similar to the particles in EXP1 the part of the plume inside the stable nocturnal boundary layer lacks all turbulent momentum keeping 5 the particles aloft and, ultimately, the particles settle down. Only these particles within the residual boundary layer above the nocturnal boundary layer remain airborne. In general, the residual layer is neutral stratified, the turbulent mixing is strongly reduced and the model uses the short approach (cf. Fig 1). Thus, the particles are trapped in this layer over night and propagate within the general air flow at almost constant altitude.

Our results above showed that the representation of diurnal cycle of the ABL is vital to an accurate trajectory calculation. 10 The here presented approach includes a TKE-based approach to account for the ABL's turbulent features. A closer look into the role of turbulence to the particle transport is provided on Fig. 7: For a reduced version of EXP2 with ~30k, we ran three different setups aiming at representing the role of the different components of the wind ultimately determining the particle's trajectory. In the first setup (Fig. 7a) we suppressed all vertical particle motion, those of the mean wind as well as those of the

turbulence. It is considered as control case. Comparable to EXP1 all particles deposit directly after the emission. Please note that the altitude is limited to 6 m.

The second setup shown in panel (b) of Fig. 7 focuses on the role of the mean wind and its ability to mix the emitted particles in the ABL. Therefore we suppressed the vertical turbulence for this setup. We observe that the mean wind of the model alone is not able to induce the particles into the atmosphere. Only seven particles (out of 30,000) get lifted up to 2000 m. Nevertheless, a glimpse on the role of the mean wind mixing to the particle trajectories can be caught. The particles perform waves with a length up to one hour and large amplitudes. The origin of these waves are the changing up and downwind regions in the convective ABL. We observed this kind of waves also in EXP2 (cf. Fig. 5c) and assign them to the mean wind. In the convective ABL, turbulence is skewed towards the upward transport since updrafts and downdrafts are not evenly distributed which may result in an underestimation of the near-surface concentration (Cassiani et al., 2015). However, we estimate the influence of this effect to be small in our model due to the high-frequency wind information.

Fig. 7c focused on the turbulent portion of the vertical mixing; the vertical mean wind is suppressed in this third setup. From this figure it becomes evident, that turbulence is essential for the entrainment of particles into the boundary layer. But it also becomes visible, that the heavier particles accumulate in the lower ABL, since the turbulent fluctuation is not as strong as the convective updrafts. Hence, it results in a not well-mixed boundary layer. In sum, the artificial setups emphasises that both components are significant for a mixing of particles from the near surface in the ABL that is conform with well-mixed criteria of Thomson (1987). Our approach provides a good balance between these components and therefore allows a realistic representation of the boundary layer dynamic in the particle trajectories.

## 4.2 Performance

COSMO use the Message Passing Interface (MPI) for parallel computing on multiple processors, which divides the model domain into subdomains with each assigned to one computing process on a singe core. Thereby, a single process exclusively holds the information of its own subdomain, so it can only calculate a particle motion that is located in its own subdomain, and if a particle leaves a subdomain, it has to be passed to the next one. To handle this, we apply the strategy proposed by Miltenberger et al. (2013): during the iteration the process gathers all particles that leave the subdomain. At the end of the time step all particles leaving the subdomain get passed to their new subdomain in a single communication step. For an even number of particles in each subdomain, the model system would be able to process them in parallel. However, when the particles travel as a plume they are most probably assign to a few or even only one subdomain. As a consequence, the calculation is performed quasi-serial. On the other hand, if we distribute the particles independent of their subdomain, we would need to pass also the atmospheric data for the calculation. This may results in a communication overhead. The efficient calculation of online trajectories remains as dilemma between communication and data availability. So further research is necessary on this technical aspect.

Our simulations were performed multiple times on different machines with 36 processors. For the simulation of EXP1 during which (nearly) all particle deposit within the first minutes, the runtime was about 40% longer than for a test case without particles. The simulation of EXP2 took roughly ten times longer than the simulation without particles. On our most recent

system (Intel(R) Xeon(R) Platinum 8160 CPUs; 2.10GHz) the runtime of EXP2 was 260 minutes compared to 36 minutes for EXP1 and 25 minutes without particles.

## 5 Conclusions

Lagrangian particle dispersion models are widely used to study the pathway of air masses and associated aerosol plumes, e.g. originating from volcanic eruptions, wildfires or desert storms. Building on the trajectory model (Miltenberger et al., 2013) online coupled to the German Weather Service's NWP model *COSMO*, we created the LPDM module *Itpas* in order to account for the turbulent nature of the convective daytime boundary layer. Here, we included the TKE on high temporal resolution, aiming for an improved representation of particle dispersion within the ABL. In order to showcase the strength of such an online coupled TKE-based approach, we simulate aerosol particle dispersion for two cases (EXP1 and EXP2) of mechanically-driven particle emission in Northeast Germany on 31 May 2017. The experiments are of particular interest as (1) the emission could be described by a source function obtained from measurements performed during the experiments; (2) particles were emitted during two different times of the day (morning and early afternoon) characterised by differently stratified boundary layers; (3) transport within the atmospheric boundary layer. In a nutshell, these conditions require to consider the air flow as non-linear and turbulent.

Simulating the particle dispersion for the emission events (EXP1 and EXP2), our results reflect the conceptual model of the boundary layer development throughout the day. Entrained into a rather stable and stratified ABL (morning hours), the particles reside only for a few minutes within the atmosphere. The bunch of trajectories launched at the source travels as a bundle with only little spreading. Finally, particles do not experience much uplift and settle down rather close to the source, within a radius of about 10 km. Compared to the morning experiment, particles entrained during the early afternoon into the convective daytime boundary layer experienced significant vertical updraft and eventually were mixed over the entire depth of the boundary layer. Travelling longer distances as remaining aloft due to turbulent updraft, the particle trajectory plume further highlights another feature of the conceptual model of the boundary layer development: the formation of the nocturnal boundary layer and the residual layer as upper part of the former daytime boundary layer above. Dust particles situated within the nocturnal boundary layer deposit due to the lack of turbulent buoyancy, whereas particles suspended within the residual layer remain aloft. Trajectories representing the first group of particles will terminate during night, whereas trajectories representing the latter group may continue travelling within the air flow.

In essence, particle entrained into a well-developed convective ABL will more likely be mixed deeper into the boundary layer, travel longer distances and reside longer within the atmosphere. This furthermore emphasises the need for the development of prognostic TKE-based LPDMs, in particular for studies on e.g. aerosol transport which source is located within the atmospheric boundary layer. The further development and application of this approach clearly would benefit from a thorough data validation such as in an extensive measurement framework.

Our approach opens up the possibility of an improved simulation of dust emissions from agricultural activities as tillage or harvest. These emissions occur mainly in weather conditions characterised by low wind speeds and affect an area several times

larger than that by wind erosion. The identification of source areas and transport paths of agricultural dust thus remains an important task, to which the presented modelling approach can make an important contribution.

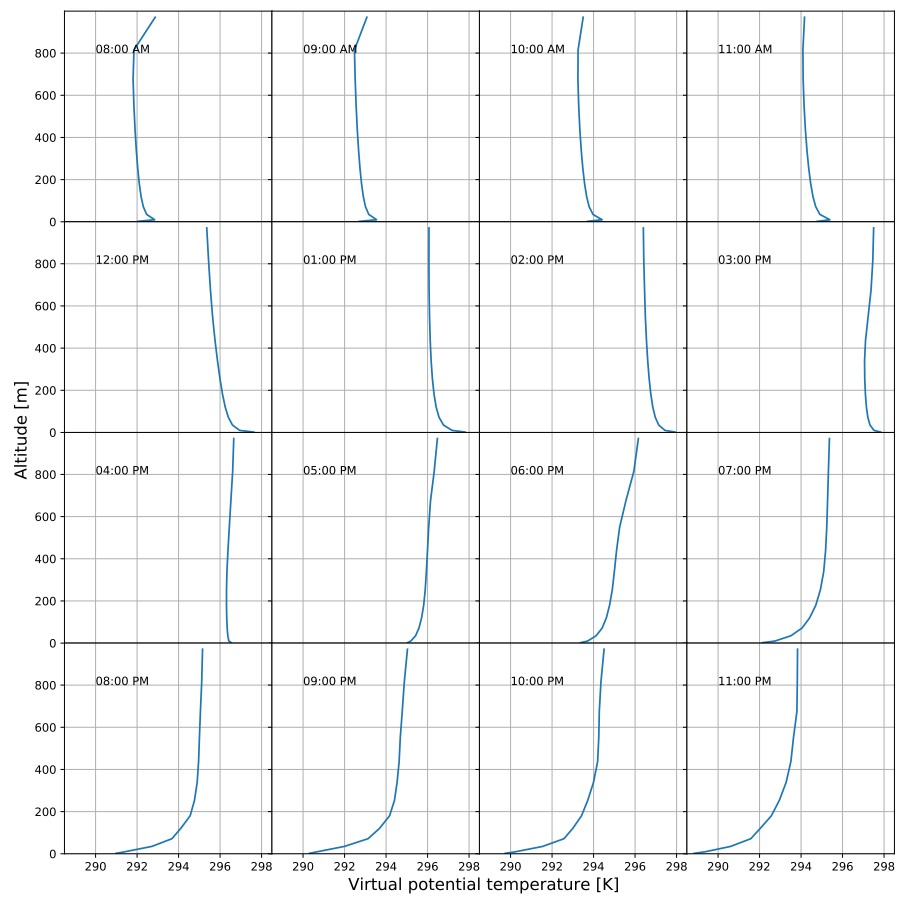

**Figure A1.** Vertical profiles of the virtual potential temperature along the mean trajectory of Fig. 6

## Appendix A

*Code and data availability.* The source code of the particle model *Itpas* is archived at Zenodo (Faust, 2020a). *Itpas* is integrated into *COSMO* and the software is restricted to owners of a COSMO license. A free license for research purposes is available from the COSMO community. http://www.cosmo-model.org/content/consortium/licencing.htm. For pre-processing, a software for creating the particle source function is freely available (Faust, 2020b). Software for processing, sorting and plotting of huge trajectory sets (Faust, 2020c) and cross-section plots along trajectories (Faust, 2020d) is open accessable. A sample data set of *Itpas'* output is available at (Faust, 2020e).

*Author contributions.* The development of the introduced model Itpas and its pre- and post-processing resources, planning and execution of the model experiments, analysing of the model results and designing of the figures were done by MF. RW suggested the model development and evaluated the model physics. The measurements were obtained and analysed by SM and RM. KS has supervised the study, suggested the model application and contributed to the analysis of the model results and the structure of the manuscript. All authors contributed to the writing and editing and reviewing processes.

*Competing interests.* The authors declare that they have no conflict of interest.

*Acknowledgements.* We are grateful to the German Weather Service (Deutscher Wetterdienst, DWD) for making *COSMO* available to us and providing of initial and boundary data. Furthermore we like to thank the two anonymous reviewers as well as P. Armand, their comments helpt to improve this study. This study was carried out in the framework of the research project "Spread of antibiotics resistance in an agrarian landscape" (SOARiAL) funded by the Leibniz Association (SAW-2017-DSMZ-2).

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
