# Peer review of "A new Lagrangian in-time particle simulation module (Itpas v1) for atmospheric particle dispersion"

_Geoscientific Model Development, 2020_

## Referee Comment (RC1) · Anonymous Referee #1 · 25 Nov 2020

Review "A new Lagrangian in-time particle simulation module (Itpas v1) for atmospheric particle dispersion"

Summary This paper reports on a new Lagrangian particle model (Itpas v1) that is designed to simulate atmospheric transport and dispersion processes for mineral dust released during farming activities. Itpas runs simultaneously with the numerical weather prediction model COSMO, thereby benefiting from high temporal resolution in the wind fields. Furthermore, the turbulence parameterisation of Itpas makes use of the prognostic turbulent kinetic energy (TKE) as calculated by COSMO. The model is applied to two field experiments. Measurements from these field experiments are used to con-

struct source functions of mineral dust particles resulting from farming activities (fertilizer spreading and tillage). The model simulates the transport and dispersion of these particles for several particle size classes. The results are discussed with respect to the virtual potential temperature and the TKE as predicted by COSMO.

General impression The paper is well-written and Figures are neat. While the use of the Langevin equation with the TKE to determine its parameters is not new, it is certainly good to see that more such models are being developed and coupled to different numerical weather prediction models. The discussion of the results with respect to the virtual potential temperature and the TKE as predicted by COSMO is an interesting test. However, I think the paper is too brief with respect to (i) context, (ii) model description and (iii) model validation. While the presented case study shows several interesting features, a thorough model validation is lacking.

Comments

1. Some context is missing about why the model is developed and how it will be used in the future. (i) why do the authors focus on Flexpart and Hysplit in the introduction? These models were constructed for continental transport, while from the case study I infer that problems of tens to a few hundreds of kilometers are of interest. (ii) transport and dispersion of mineral dust is mentioned as an application; could the authors provide an overview of the current state-of-the-art in that field? And how does this approach contribute in light of the current state-of-the-art? Are the results different than what one would get when using Flexpart, or a Gaussian puff model? (iii) How will the output of Itpas be used? Is a deposition map of dust particles the final goal as in Section 3? In the presented test case, one model particle represents 5 million physical particles; will that be sufficient for operational/research use of Itpas (that is, what minimum concentration levels are still relevant)?

2. Is there a particular reason for coupling Itpas to COSMO, since I understood that COSMO will be replaced by ICON in the near future?

3. While I appreciate the clear description of the model physics in Section 2.2, I think that some additional information is needed about the model: (i) Could the authors give an idea about the model's computational requirements: how long does the calculations take and on what type of machine? (ii) The paper does not mention if and what parameters need to be specified before running the model. Could the authors describe these parameters, their default value and the impact of perturbing these values?

4. Section 3.2 "Source function": I think this is an interesting approach to determine the source function. Could the authors comment on the added value of this elaborated approach (or: what would the result be if a simpler approach would be used), especially when there is long-range transport?

5. EXP1: While it makes sense from Fig 6 that the stable conditions prohibit plume rise, and consequently a lot of particles are deposited within the first kilometers, I wonder to what extend the results presented in Fig 4 are truly physical: first, if the true transport is indeed limited to only 10 km, then Itpas (driven with meteorological data with grid spacings of 2.8 km) does not seem appropriate to model this. (ii) I'm surprised that the bulk of the plume remains below 5 m: I would expect that mechanical turbulence (due to buildings, a forest) could force the plume to higher elevations. Is the effect of mechanical turbulence taken into account, and if so, how?

6. Figure 5 (a): Could the authors comment on why the horizontal dispersion depends on particle size? In particular because the spread seems symmetric around the mean path of the plume (so that a difference due to vertical wind shear can be excluded, which would show an asymmetric spread around the mean path of the plume).

7. Section 3: While the authors describe two field experiments for which they apply Itpas, no formal model validation is performed (the measurements are not used to validate the model output but to construct source function). I think the lack of such test cases is an important drawback for this study.

Minor comments

p 1, line 6: "approximation": I would suggest to be more specific since all models use approximations at some point

p 2, line 9: "uncertainties due to turbulence" → I suggest "deviations due to turbulence"

p 2, line 20: what is the time step of Itpas? Since it uses NWP data with higher resolution than considered in Seibert (1993), the time step should be much smaller than a few minutes?

p 2, line 24: I suggest to add some motivation for avoiding temporal interpolation

p 3, line 19: "There is only a weak link between Itpas and COSMO." Could Itpas be used with data from other NWP models?

p 3, line 27: why is the forward Euler method used? It it considered sufficiently accurate because the turbulent velocity is expected to fluctuate around 0?

p 3, line 28: "Since the particle is not allowed to leave the boundary layer directly, the model checks if the particle is still inside the boundary layer after the motion.": (i) how is the boundary layer height determined? (ii) In contrast, Verreyken et al (2019) use the TKE to allow "novel turbulent modes [...] to mix boundary layer air with free-tropospheric air masses". (iii) Particles can only leave the boundary layer via the re-solved mean wind?

Verreyken, B., Brioude, J., & Evan, S. (2019). Development of turbulent scheme in the FLEXPART-AROME v1.2.1 Lagrangian particle dispersion model. Geoscientific Model Development, 12(10), 4245-4259.

Figure 1: There should be an option "No" starting from the question "at surface?" that leads to "particle alive".

p 4, line 1: what is the motivation for the reduced approach? I can think of a reduction in calculation time, but that might not matter too match if only few particles move above the boundary layer.

p 5, line 23: The Lagrangian time scale tau_L is a vector (dependent on the spatial direction), so that the expression dt » tau_L is incorrect. I assume it should be valid for each spatial direction, so that dt » tau_L_i with i ={x,y,z}.

p 5, Eqs (3) and (4): some additional information or a reference would be welcome - why is it defined the way it is and what are the underlying assumptions and implications?

p 5, line 26: do the authors implicitly assume Gaussian turbulence?

p 6, line 9: "t" is not defined here (unless "dt" was meant)

p 6, Eq (9): "sigma_u" should be "sigma" (omit the subscript)

p 7: Eq(13): brackets are missing after "exp"

p 8, line 5: I suggest to replace "vertically" by "perpendicular"

p 9, line 10: "conservative": while this is conservative with respect to the released number of particles, it is not conservative with respect to the impact (underestimating the source = underestimating the impact)

p 13, Figure 6: the mean trajectory and the standard deviation suggest that the bulk of the plume is lifted from the ground. I would expect instead that the plume is mixed homogeneously between the surface and the top of the plume (which is also what Figure 5, c, suggests).
* * *

---

## Referee Comment (RC2) · Anonymous Referee #2 · 3 Dec 2020

The authors presented new Lagrangian particle dispersion model (Itpas) for particle transport within a boundary layer which can be, possibly, turbolent. The model is online coupled with weather forcast model COSMO (German Weather Servise). The Itpas model is applied to two fields experiments studing the behavior of particles released by the agricultural activities (fertilization and cultivation with tractor). The paper is well writen and clear to understand, however, my main concern is regarding the validation of the Itpas model, see comments.

Specific comments:

1. p.3 l.31 and p.5 l.16: please, specify the probability function, is it normal distribution
with mean 0 and variance 1? Could you please, briefly verify this choice?

2. p.9 l.10: I am quite confused by the assumption that "the particle concentration becomes zero at a height of 5 m". Is this realistic? Could you, please, discuss this choice? Maybe, it can be seen from photos.

3. I am not sure about the role of two measurement points mentioned in the Experiment part. Are the roles of these points only to concstruct the source function? If this is the case, than I do not see the merit of the simulation experiment in Sec. 3.3 regarding validation of the Itpas model itself (although the simulation itself is interesting with discussion on Fig. 6). However, this means that the Itpas model is not validated in the paper. Please, clarify.

Minor comments:

1. Eq. (5): u, v, and w are probably spatial directions but it should be stated in the text.

2. Eq. (17): $erf$ should be probably $erf(\gamma)$.

3. p.15 l.20: crating –> creating

4. Reference (Pisso et al., 2019) is already published, please, update the citation.

---

## Referee Comment (RC3) · P. Armand (Referee) · 16 Dec 2020

The objectives of this paper are to present and to exemplify a new Lagrangian Particle Dispersion Model (LPDM) called "Itpas". As the other models in this category, Itpas computes multiple individual trajectories of air parcels (possibly carrying pollutants). The air parcels are transported by the mean components of the wind velocity and dispersed by the fluctuating components of the wind velocity. The velocity fluctuations are due to the turbulence and represented as random deviations from mean trajectories. The fluctuating components of the wind verify the Langevin equation and are modelled according to Thomson proposals.

The parameters of the LPDM are the standard deviations of the wind velocity components and the Lagrangian time scales, which can be related to the Turbulent Kinetic Energy (TKE) of the flow. As argued by the authors, the distinctiveness of Itpas is to use the high-frequent wind information and the prognostically calculated TKE issued by the German Weather Service's mesoscale weather forecast model COSMO. Moreover, Itpas is coupled on-line with COSMO.

The authors give an example of application of the COSMO-Itpas modelling chain for a case-study of agricultural solid particle emission in Eastern Germany. The simulation results regarding horizontal and vertical transport and dispersion of the particles are discussed with regards to the circadian evolution of the turbulent Atmospheric Boundary Layer (ABL). As underlined by the authors, the results suggest that the Itpas model represent correctly and quite accurately the transport and dispersion of the emitted agricultural particles.

The paper is well-written and well-structured. It is interesting and worth being published. I have some remarks and questions for the authors, which should be answered before the publication of the paper.

Page 3 - Line 9 - Is it possible to use Itpas both on-line and off-line using either COSMO weather forecasts or COSMO weather analyses? Can the authors comment on the applicability of these two approaches?

Page 5 - Line 27 - I wonder if COSMO can provide only the TKE (and horizontal and vertical diffusion coefficients) or if the meteorological model could issue the standard deviations of the three velocity components? For the LPDM, the anisotropic fluctuating components of the velocity would be much more interesting than the TKE.

Page 6 - Line 3 - The "m_i" factors describe the weighting of the TKE in the three spatial directions. The values of these factors depend on the stratification conditions in the ABL. Could the authors explain in more details how the "m_i" factors are related to the components of the mean wind?

[Figure]

Page 6 - Line 16 - I don't see any major difference between the Itpas model and the FLEXPART or HYSPLIT models. Could the authors comment on discrepancies, if any, between Itpas and these models?

Page 9 - Line 6 - I would not say that the flow conditions above the ABL are nearly laminar. The authors should consider revising this sentence.

Page 9 - Figure 3 - I wonder if the source term modelling depicted in Figure 3 applies for both EXP1 and EXP2. This is not clearly mentioned in the paper. Can the authors clarify this point?

Page 9 - Line 19 - The number of numerical particles (100,000 in EXP1 and 270,000 in EXP2) used in the Itpas simulations seems to me quite low. I wonder if this number is enough at least in EXP2 with particles supposed to travel several hundreds of kilometers. Was a sensitivity study about the number of numerical particles carried out by the authors?

Page 10 - Line 5 - The source term in EXP1 and EXP2 is contained in a volume of about 10x10x5 m3. The initial distribution of the numerical particles in this volume is given by a detailed model. Is it really necessary to be so precise (see Figure 3) when considering the horizontal and vertical dimensions of the simulation domain for the atmospheric transport and dispersion?

Page 11 - Figure 4 - In EXP1, it is not clear for me if the particles are more or less lifted depending on their diameters. Can the authors clarify this point?

Page 11 - Figure 4 - Moreover, I'm concerned how the meteorological model can give information so close to the ground and along a so short horizontal distance (10 to 20 km) and height (5 m). As for me, there is an inconsistency between the space and time resolution of COSMO meso-scale wetaher forecast and the micro-scale transport and dispersion of the particles in EXP1. Explanations and justification from the authors are needed here! (I'm more confident with EXP2 simulation.)

[Figure]

Page 12 - Figure 5 - I'm a bit surprised by the large horizontal expansion of the particles plume in Figure 5a even if explanations are given by the authors in the paper (development of the ABL and turbulent diffusion around noon - see Page 14 - Line 9). I'm even more surprised by the vertical ascending motion not of the smallest particles (less than 1 or 2 $\mu$m), but of the largest particles (up to 30 $\mu$m). Looking at Figure 5c, it seems that there are only very few differences between the aerodynamic behavior of the smallest and largest solid particles. Can the authors comment on this point?

Page 12 - Figure 6 - This is not easy at all to figure out the gradient of the virtual potential temperature just by looking at Figure 6. It would be simpler to determine the stratification of the ABL by visualizing vertical profiles of the temprature gradient graphed at successive hours of the day. I suggest the authors to add this information in supplementary materials.

Page 14 - Line 4 - What is supposed to be evident is actually not so obvious. See my remark just before.

Page 14 - Line 19 - What are the computational times to simulate EXP1 and EXP2? Would it be possible to use Itpas off-line? What would be the difference in the computational times?

Page 15 - Line 12 - Most of LPDM dedicated to local or regional scale simulations use the TKE to evaluate the variances of the velocity fluctuations and the Lagrangian time scales in the three spatial directions. This is probably less the case of LPDM adapted to larger scales like FLEXPART or HYSPLIT. The difficult point with TKE remains to distribute the turbulence between the space directions. Could be the authors recall how they proceed in Itpas (see my previous question about Page 6 - Line 3) and comment on this aspect?

Page 15 - Line 16 - According to the authors, the area affected by agricultural emissions are "several times larger" than by wind erosion. This is not obvious. I wonder why?

---

## Author Comment (AC1) · 4 Feb 2021

**1 Review #1**

We would like to thank the Editor and the reviewer for their time spent on the manuscript and the comments and suggestions made. We carefully considered all of them; they helped us to improve the manuscript. Please find below the point-by-point reply with reviewer's comments printed in italics. Authors' comments are given below the reviewer's comment.

*Summary This paper reports on a new Lagrangian particle model (Itpas v1) that is designed to simulate atmospheric transport and dispersion processes for mineral dust released during farming activities. Itpas runs simultaneously with the numerical weather prediction model COSMO, thereby benefiting from high temporal resolution in the wind fields. Furthermore, the turbulence parameterisation of Itpas makes use of the prognostic turbulent kinetic energy (TKE) as calculated by COSMO. The model is applied to two field experiments. Measurements from these field experiments are used to construct source functions of mineral dust particles resulting from farming activities (fertilizer spreading and tillage). The model simulates the transport and dispersion of these particles for several particle size classes. The results are discussed with respect to the virtual potential temperature and the TKE as predicted by COSMO.*

*General impression The paper is well-written and Figures are neat. While the use of the Langevin equation with the TKE to determine its parameters is not new, it is certainly good to see that more such models are being developed and coupled to different numerical weather prediction models. The discussion of the results with respect to the virtual potential temperature and the TKE as predicted by COSMO is an interesting test. However, I think the paper is too brief with respect to (i) context, (ii) model description and (iii) model validation. While the presented case study shows several interesting features, a thorough model validation is lacking.*

Many thanks for this encouraging comment. We have taken on your comment regarding the degree of detail of the manuscript's content and extended the corresponding paragraphs regarding context, model description and model validation. Please see below for further details on changes made.

**1.1 General comments**

*1. Some context is missing about why the model is developed and how it will be used in the future. (i) why do the authors focus on Flexpart and Hysplit in the introduction? These models were constructed for continental transport, while from the case study I infer that problems of tens to a few hundreds of kilometers are of interest. (ii) transport and dispersion of mineral dust is mentioned as an application; could the authors provide an overview of the current state-of-the-art in that field? And how does this approach contribute in light of the current state-of-the-art? Are the results different than what one would get when using Flexpart, or a Gaussian puff model? (iii) How will the output of Itpas be used? Is a deposition map of dust particles the final goal as in Section 3? In the presented test case, one model particle represents 5 million physical particles; will that be sufficient for operational/research use of Itpas (that is, what minimum concentration levels are still relevant)?*

Many thanks for this detailed comment, which we would like to address following the structure of your comment. (i) Our intention behind starting the introduction section with FLEXPART and HYSPLIT is that readers may not be familiar with the general concept of a Lagrangian particle dispersion model, however, FLEXPART and HYSPLIT are well-known and widely applied models in the broader atmospheric research community. By referring to a generally known model system, we aim at providing the reader with a rough idea on the direction the manuscript is heading thematically before we get more specific on the explicit outcome of the presented work.

(ii) Thanks for pointing this out to us. For a better understanding of Itpas' applicability, we have added a section presenting a brief overview on dust modelling in general and associated challenges to the work presented in order to provide a broader context for this study. FLEXPART is designed for global and mesoscale applications. For applications at smaller scales, there are models like microSPRAY for local-scale phenomena

such as dispersion in street canyons; these models are often driven by LES data. With Itpas we are somewhere in between the mesoscale and local-scale application. We designed our model for particle transport ranges that are too long to be captured by the (local) micro-scale models, but at the same time for particle sources that will likely be underrepresented in mesoscale model settings.

(iii) Generally, there are several options to use the Itpas model output. In order to illustrate the range of possibilities, we would like to first lay out how we use the model output in our research. In the framework of the research study, Itpas was developed, we focus on the vertical spreading of the trajectories. This allows for retrieving information on how well the model performs regarding mixing particles vertically over the depth of the boundary layer. This is a key performance measure for this kind of model. As the model outputs the trajectory data for every individual particle, various model applications are reasonable as different kind of parameters can be retrieved. For a study focusing on aerosol particles, the entire path from source to sink may be of interest. For studies of air quality, the trajectory density might be considered in more detail. And for studies of the airflow, the particle trajectory itself may provide the most relevant information. Ultimately, the aim of this study is to illustrate possibilities using the trajectory data creatively - by itself and in combination with other atmospheric data. All post-process examples are freely available on GitHub.

*2. Is there a particular reason for coupling Itpas to COSMO, since I understood that COSMO will be replaced by ICON in the near future?*

There is no particular reason in the sense that Itpas can only be coupled to COSMO. Despite the ICON development and COSMO's retirement as an operational weather forecast model, COSMO still serves as a state of the science atmospheric research model which is used by a broad scientific community.

[Figure]

*3. While I appreciate the clear description of the model physics in Section 2.2, I think that some additional information is needed about the model: (i) Could the authors give an idea about the model's computational requirements: how long does the calculations take and on what type of machine? (ii) The paper does not mention if and what parameters need to be specified before running the model. Could the authors describe these parameters, their default value and the impact of perturbing these values?*

Many thanks for your question. Regarding (i), the calculation time depends on the number of particles and how long they remain in the atmosphere. But also on how strong the particles are dispersed in the model domain. We did not perform specific experiments on the calculation time; we run our simulations on different servers with 36 processors. The actual calculation time depends on the processor generation, which was different for the different servers. However, we found that the simulation of EXP2 with 270k particles roughly takes 10 times longer than the simulation without particles. On our most modern machine (Intel(R) Xeon(R) Platinum 8160 CPU; 2.10GHz) that were 25 minutes for the experiment without and 260 minutes for the experiments with the particles. The simulation of EXP1, where most particles fall down directly, took only 40% longer than the no particle case.

(ii) A description of how to set up the model is included in the README-file of the model that we now added to the supplement.

*4. Section 3.2 "Source function": I think this is an interesting approach to determine the source function. Could the authors comment on the added value of this elaborated approach (or: what would the result be if a simpler approach would be used), especially when there is long-range transport?*

With the for this application developed source term, we aim at representing the initial

dust plume emitted by the tractor's tool as accurate as possible. The benefit of this method becomes more visible when discussing possible simplifications. The most simple approach would be to use two point sources (one for each measurement height) and use the observed particle number as input. Hereby, we would miss the particles underneath, between and above the inlets of the measurement device. This points towards the necessity of a vertical profile. But by only considering one profile we do not fully represent the particle number of the entire plume. Consequently, we need to define the volume of the particle plume. This brought us to our 3D bubble approach as presented here.

*5. EXP1: While it makes sense from Fig 6 that the stable conditions prohibit plume rise, and consequently a lot of particles are deposited within the first kilometers, I wonder to what extend the results presented in Fig 4 are truly physical: first, if the true transport is indeed limited to only 10 km, then ltpas (driven with meteorological data with grid spacings of 2.8 km) does not seem appropriate to model this. (ii) I'm surprised that the bulk of the plume remains below 5 m: I would expect that mechanical turbulence (due to buildings, a forest) could force the plume to higher elevations. Is the effect of mechanical turbulence taken into account, and if so, how?*

Many thanks for pointing this out. With regard to the first part of your comment (i), indeed, the horizontal resolution is not sufficient for such a short transport range. We also did not expect this result beforehand and we would like to explain how we think the results shall be understood. EXP1 should be considered as no transport case because the particles fall down directly after emission and the only precondition that allows for the calculated transport is the fact that small particles have small settling velocities.

(ii) The answer to this question has two aspects to consider. First, the model does not take into account small surface structures like trees, buildings or forests that could force a vertical movement of the plume. If the dust plume would pass a forest or an

urban area it may slow down because of a reduced wind velocity in the COSMO model. But the particles would pass the area without being captured or lifted. Nevertheless, mechanical lifting is included in the model to account for changes in orography. However, this effect is not evident from our figures because there the relative height above the surface is shown.

*6. Figure 5 (a): Could the authors comment on why the horizontal dispersion depends on particle size? In particular because the spread seems symmetric around the mean path of the plume (so that a difference due to vertical wind shear can be excluded, which would show an asymmetric spread around the mean path of the plume).*

Of course. This effect comes from the fact that there are way more small particles in the simulation than larger ones. With increasing particle number, the chances for outliers increase.

*7. Section 3: While the authors describe two field experiments for which they apply Itpas, no formal model validation is performed (the measurements are not used to validate the model output but to construct source function). I think the lack of such test cases is an important drawback for this study.*

The manuscript aims at addressing two aspects of aerosol dispersion modelling, here discussed for dust particle emission from arable land driven by mechanical soil preparation: (1) The presentation of a dust emission function that reflects the artificial (mechanically driven) nature of the dust entrainment process - other than aeolian, wind-driven emission for which the emission is a function of the wind speed. (2) The representation of the dust particles trajectory through the atmosphere. For the latter motivation, in particular, the model's ability to reflect the turbulent nature of the boundary layer

off-setting the mean travel path of an airborne particle is focused. Other than for wind-driven soil erosion, where emission fluxes are calculated as a function of wind speed, mechanically driven dust entrainment has up to our knowledge not been parameterised explicitly so far. As vital to the accurate representation of particle trajectories, we determine a source function from particle measurements obtained during the entrainment process (bypassing tractor pulling a tool disturbing the soil). Used as input information determining the particle source function, these data cannot be used for validation anymore. As an accurate as possible source function is essential to the calculation of the aerosol dispersion, and the measurements at source were not suitable for validating the transport anyway due to its too short distance, we decided to use the measurement data to obtain the best estimate of the particle entrainment. As said above, getting the source term right is essential to the correct representation of the particles' trajectories. This is followed by an as accurate as possible representation of the atmospheric dynamics, which ultimately determine the trajectories' pathway. To validate this thoroughly, a specific to this needs designed measurement setup would be desirable. In particular, having particle sampler at different height and different distances from the source in order to estimate concentration gradients and transport altitudes. As these data are currently not available, we wish to get the opportunity to thoroughly validate the here presented model system in future. Comparing against satellite retrieved aerosol concentrations fails due to too low aerosol concentrations. The second focus of our study is on evaluating the dynamics of the ABL, in particular the impact of its diurnal cycle on dust aerosol dispersion. The NWP model COSMO is well established and thus very likely able to represent correctly the atmospheric dynamics. Our particle trajectory model bases on well-known approaches too: we use the Petterson scheme for online trajectory calculation (Miltenberger et al. 2013) and the turbulent fluctuations for the LPDM (Hall 1975). In concert, known and established model components are used in our approach. The novelty we added to this model-setting is the use of the high-frequency TKE information to trigger the turbulent fluctuations on the mean wind component. Having the "ingredients" to an accurate trajectory calculation in mind, the

best chance to evaluate particle motion is the behaviour of the particles in the mixed ABL. Up to now, and based on our results, we can say that the particles do what we would expect from the theory. In a nutshell: The particles are well mixed over the depth of the boundary layer, we do not have accumulations at the surface or at the ABL's top, and we have no overshooting above the ABL and no drastic loss at the surface.

**1.2   Minor comments**

*p 1, line 6: "approximation": I would suggest to be more specific since all models use approximations at some point*

We appreciate the reviewers suggestion and understand its argumentation, however, we nevertheless prefer to use the term approximation here to specifically underline that the widely common approach is to bypass an explicit parameterisation.

*p 2, line 9: "uncertainties due to turbulence" > I suggest "deviations due to turbulence"*

Done

*p 2, line 20: what is the time step of Itpas? Since it uses NWP data with higher resolution than considered in Seibert (1993), the time step should be much smaller than a few minutes?*

The Itpas time step is the time step of the forecast simulation. We recommend using Itpas for simulations with a high spatial resolution (below 0.1°). For such simulations, the time step is, in general, less than 60 seconds.

[Figure]

*p 2, line 24: I suggest to add some motivation for avoiding temporal interpolation*

On page 2 line 23 we state "This is a crucial issue, especially for small-scale applications at short time scales."

*p 3, line 19: "There is only a weak link between Itpas and COSMO." Could Itpas be used with data from other NWP models?*

Yes, this is one big advantage of the "weak link" between Ipas and COSMO. Itpas could also be coupled to other NWP models because it has its own I/O system and 'only' need the data stream of the wind, turbulence and a view other variables and a call statement each model time step. Currently, the structure of Itpas matches with the structure of COSMO though.

*p 3, line 27: why is the forward Euler method used? It considered sufficiently accurate because the turbulent velocity is expected to fluctuate around 0?*

For NWP models, the turbulent wind is completely random. Thus, calculating the new position directly or via iteration provides similar results. In essence, the turbulent wind component remains a random jump in a random direction. Additionally, the turbulent part of the motion is for most cases significantly smaller than the mean part of the motion. So the possible error of the simple integration is comparably low. For the mean wind motion, we use an accurate integration scheme.

*p 3, line 28: "Since the particle is not allowed to leave the boundary layer directly, the model checks if the particle is still inside the boundary layer after the motion.":*

*(i) how is the boundary layer height determined? (ii) In contrast, Verreyken et al (2019) use the TKE to allow "novel turbulent modes [...] to mix boundary layer air with freetropospheric air masses". (iii) Particles can only leave the boundary layer via the resolved mean wind? Verreyken, B., Brioude, J., & Evan, S. (2019). Development of turbulent scheme in the FLEXPART-AROME v1.2.1 Lagrangian particle dispersion model. Geoscientific Model Development, 12(10), 4245-4259.*

(i) We use the boundary layer height that is provided by COSMO. In COSMO, the boundary layer height is determined with the bulk Richardson method, which describes the depth of the boundary layer as the height where the bulk Richardson number reaches a critical value, which is set to 0.33 for stable and 0.22 for unstable conditions.

(ii) Many thanks for pointing us towards this work. We absolutely agree, turbulent mixing at the upper edge of the boundary layer and thus entrainment of air into the free troposphere is an interesting question, which we will consider for future development.

(iii) In the current model version, particles are somewhat trapped in the boundary layer. Also, particles that were lifted by the mean wind through the boundary layer's top will be reflected at the layer's top. However, particles can leave the boundary layer via orographically lifting or when the ABL collapses (e.g. after sunset) around the particles. In the latter case, the particles are then situated within the residual layer above the top of the newly forming nocturnal boundary layer.

*Figure 1: There should be an option "No" starting from the question "at surface?" that leads to "particle alive".*

Yes, thanks, changed.

*p 4, line 1: what is the motivation for the reduced approach? I can think of a reduction in calculation time, but that might not matter too match if only few particles move above the boundary layer.*

During the night time, most of the particles are above the shallow nocturnal boundary layer within the residual layer. During these hours, the calculation can be faster.

*p 5, line 23: The Lagrangian time scale tau_L is a vector (dependent on the spatial direction), so that the expression dt Âż tau_L is incorrect. I assume it should be valid for each spatial direction, so that dt Âż tau_L_i with i ={x,y,z}.*

Done

*p 5, Eqs (3) and (4): some additional information or a reference would be welcome - why is it defined the way it is and what are the underlying assumptions and implications?*

The reference for this is Hall (1975) which is already included. Indeed, it is not obvious for the reader that this reference corresponds to the equations (2 - 4). This will be clarified.

*p 5, line 26: do the authors implicitly assume Gaussian turbulence?*

In the LPDM, the turbulent fluctuation is defined as:

$$\vec{u}'_{t+1} = \vec{a}\,\vec{u}'_t + \vec{b}\,\vec{\xi}$$

where the first term includes the current direction of the turbulence and the second term adds random noise to the system. Since the random number $\vec{\xi}$ is taken from a Gaussian distribution the turbulent fluctuation appears in this shape as well.

*p 6, line 9: "t" is not defined here (unless "dt" was meant)*

Yes, t was meant, done

*p 6, Eq (9): "sigma_u" should be "sigma" (omit the subscript)*

Done

*p 7: Eq(13): brackets are missing after "exp"*

Done

*p 8, line 5: I suggest to replace "vertically" by "perpendicular"*

Done

*p 9, line 10: "conservative": while this is conservative with respect to the released number of particles, it is not conservative with respect to the impact (underestimating the source = underestimating the impact)*

We agree, underestimating the strength of an aerosol source may ultimately result in

an underestimation of the aerosol impact. In this case here, we are confident with our approach as these particles close to the surface may deposit rather sooner than later.

*p 13, Figure 6: the mean trajectory and the standard deviation suggest that the bulk of the plume is lifted from the ground. I would expect instead that the plume is mixed homogeneously between the surface and the top of the plume (which is also what Figure 5, c, suggests).*

Around noon and during the early afternoon, dust is mixed homogeneously over the depth of the boundary layer due to convective mixing. In the evening after sunset, the plume appears lifted above the ground as described in the manuscript. However, the illustration of the mean trajectory has the weakness as it is defined as the mean of the trajectories that reach the eastern edge of the domain. As a consequence of this definition, particles that tend to travel on lower altitude, and may also deposit earlier, are excluded. Thus the mean trajectory with its standard deviations may appear located at higher altitudes.

**References**

Hall, C. D.: The simulation of particle motion in the atmosphere by a numerical random-walk model, Quarterly Journal of the Royal Meteorological Society, 101, 235–244, https://doi.org/10.1002/qj.49710142807, 1975.

Miltenberger, A. K., Pfahl, S., and Wernli, H.: An online trajectory module (version 1.0) for the nonhydrostatic numerical weather prediction model COSMO, Geoscientific Model Development, 6, 1989–2004, https://doi.org/10.5194/gmd-6-1989-2013, 2013.

---

## Author Comment (AC2) · 4 Feb 2021

**1 Review #2**

We would like to thank the Editor and the reviewer for their time spent on the manuscript and the comments and suggestions made. We carefully considered all of them; they helped us to improve the manuscript. Please find below the point-by-point reply with reviewer's comments printed in italics. Authors' comments are given below the reviewer's comment.

*The authors presented new Lagrangian particle dispersion model (Itpas) for particle transport within a boundary layer which can be, possibly, turbulent. The model is online coupled with weather forcast model COSMO (German Weather Servise). The Itpas model is applied to two fields experiments studing the behavior of particles released by the agricultural activities (fertilization and cultivation with tractor). The paper is well writen and clear to understand, however, my main concern is regarding the validation of the Itpas model, see comments.*

1.1  General comments

*1. p.3 l.31 and p.5 l.16: please, specify the probability function, is it normal distribution with mean 0 and variance 1? Could you please, briefly verify this choice?*

Many thanks for this comment. Two different things were mentioned here. Regarding the first part: p.3 l.31 refers to the probability function of the dry deposition (eq (16)). We have added a reference to clarify this. Regarding the second part: p.5 l.16 refers to the random number of the Lagrangian process( $\xi$). This is indeed taken from a normal distribution with a mean value of zero and a standard deviation of 1. We have added this information to the manuscript. The random number deflects the trajectory from its current path. As this process has to be symmetric, the mean value of the random number has to be zero. The magnitude of the disturbance is the standard deviation of the wind $\sigma$ multiplied by the random number. With the chosen random number distribution we then estimate the turbulent disturbance that is most likely (68%) as this is the behaviour needed here.

*2. p.9 l.10: I am quite confused by the assumption that "the particle concentration becomes zero at a height of 5 m". Is this realistic? Could you, please, discuss this choice? Maybe, it can be seen from photos.*

The source function describes the initial state of the particles before they start to travel. Here, dust particles were uplifted mechanically by the tool pulled by a tractor. Once airborne, the particles immediately start to disperse with the ambient airflow so that the process of the initial emission transforms seamlessly into the process of transport. So the upper edge of the emitting plume cannot clearly be defined. However, with regard to the model application, we need to define an initial state of the particles as starting point. The measurements show that there was a reasonable concentration evident at 3.8m, so we defined 5m as the upper edge of the emission plume.

*3. I am not sure about the role of two measurement points mentioned in the Experiment part. Are the roles of these points only to concstruct the source function? If this is the case, than I do not see the merit of the simulation experiment in Sec. 3.3 regarding validation of the Itpas model itself (although the simulation itself is interesting with discussion on Fig. 6). However, this means that the Itpas model is not validated in the paper. Please, clarify.*

The measurements are used to construct the source plume. The two vertically stacked measurement points allowed us to define an idealised particle plume behind the tractor that represents the measured particle number for different particle size ranges. This itself is (as far as we know) a novel approach to define this kind of particle source. The measurements are indeed not used for validation, mainly because we cannot validate the model's source function with the data that we used for determining the source function. Additionally, the data are not suitable for validating particle transport as the measurements are too close to the source and therefore do not reflect any occurring particle transport.

1.2   Minor comments

*1. Eq. (5): u, v, and w are probably spatial directions but it should be stated in the text.*

Done

*2. Eq. (17): $erf$ should be probably $erf(\gamma)$.*

Done

*3. p.15 l.20: crating –> creating*

Done

*4. Reference (Pisso et al., 2019) is already published, please, update the citation.*

Done

---

## Author Comment (AC3) · 4 Feb 2021

**1    Review #3**

We would like to thank the Editor and the reviewer for their time spent on the manuscript and the comments and suggestions made. We carefully considered all of them; they helped us to improve the manuscript. Please find below the point-by-point reply with reviewer's comments printed in italics. Authors' comments are given below the reviewer's comment.

*The objectives of this paper are to present and to exemplify a new Lagrangian Particle Dispersion Model (LPDM) called "Itpas". As the other models in this category, Itpas computes multiple individual trajectories of air parcels (possibly carrying pollutants). The air parcels are transported by the mean components of the wind velocity and dispersed by the fluctuating components of the wind velocity. The velocity fluctuations are due to the turbulence and represented as random deviations from mean trajectories. The fluctuating components of the wind verify the Langevin equation and are modelled according to Thomson proposals.*

*The parameters of the LPDM are the standard deviations of the wind velocity components and the Lagrangian time scales, which can be related to the Turbulent Kinetic Energy (TKE) of the flow. As argued by the authors, the distinctiveness of Itpas is to use the high-frequent wind information and the prognostically calculated TKE issued by the German Weather Service's mesoscale weather forecast model COSMO. Moreover, Itpas is coupled on-line with COSMO. The authors give an example of application of the COSMO-Itpas modelling chain for a case-study of agricultural solid particle emission in Eastern Germany. The simulation results regarding horizontal and vertical transport and dispersion of the particles are discussed with regards to the circadian evolution of the turbulent Atmospheric Boundary Layer (ABL). As underlined by the authors, the results suggest that the Itpas model represent correctly and quite accurately the transport and dispersion of the emitted agricultural particles. The paper is well-written and well-structured. It is interesting and worth being published. I have some remarks and questions for the authors, which should be answered before the publication of the paper.*

Many thanks for your encouraging comment. We have carefully considered your comments made below and provided answers to your questions.

**1.1 General comments**

*Page 3 - Line 9 - Is it possible to use ltpas both on-line and off-line using either COSMO weather forecasts or COSMO weather analyses? Can the authors comment on the applicability of these two approaches?*

Currently, the model runs as on-line application only. In an off-line setting, one crucial issue arising is the interpolation in time between the available input data. Depending on the input model this time increment can be up to six hours, depending on the data set used (e.g. some reanalysis data are available at 6-hourly resolution). But this is critical for applications with a time scale of only a view hours. In our research framework, we aim at model-based studies at a regional scale in space (a few hundreds of kilometres) and short temporal scale. In such a setup the simulation benefits from the online system because the temporal interpolation and the associated error can be avoided.

*Page 5 - Line 27 - I wonder if COSMO can provide only the TKE (and horizontal and vertical diffusion coefficients) or if the meteorological model could issue the standard deviations of the three velocity components? For the LPDM, the anisotropic fluctuating components of the velocity would be much more interesting than the TKE.*

Since we are calculating a particle motion for every time step of the NWP model, anisotropic fluctuations are included in our particle motion. The TKE induced turbulence is put on top to create the particle dispersion.

*Page 6 - Line 3 - The "m_i" factors describe the weighting of the TKE in the three spatial directions. The values of these factors depend on the stratification conditions in*

*the ABL. Could the authors explain in more details how the "m_i" factors are related to the components of the mean wind?*

*Page 15 - Line 12 - Most of LPDM dedicated to local or regional scale simulations use the TKE to evaluate the variances of the velocity fluctuations and the Lagrangian time scales in the three spatial directions. This is probably less the case of LPDM adapted to larger scales like FLEXPART or HYSPLIT. The difficult point with TKE remains to distribute the turbulence between the space directions. Could be the authors recall how they proceed in Itpas (see my previous question about Page 6 - Line 3) and comment on this aspect?*

We as the two comments above refer to the same topic, we chose to address them at once. We thank you to pointing this out. Indeed, the distribution of the TKE is an important attribute of the model and we have addressed briefly only. We provide more details on this in the revised version of the manuscript.

We distribute the TKE with the fractions of the mean wind, whereby the mean wind in the model fluctuates with every time step. So, in stable conditions for which the TKE is low, the vertical turbulent fluctuation is suppressed. In unstable conditions when the TKE and the mean vertical wind increases, we see stronger vertical turbulent fluctuations. The combination of the fluctuating mean wind and the vertical turbulence then produces the vertical mixing that we can observe for EXP2. For a better understanding of the interaction between mean and turbulent wind components, we added an additional figure (Fig. 7) to our discussion.

*Page 6 - Line 16 - I don't see any major difference between the Itpas model and the FLEXPART or HYSPLIT models. Could the authors comment on discrepancies, if any, between Itpas and these models?*

The main difference to FLEXPART and HYSPLIT is, that we use the prognostic TKE from the driving NWP model for our calculations. Thus, we see our model being applicable to smaller-scaled processes.

*Page 9 - Line 6 - I would not say that the flow conditions above the ABL are nearly laminar. The authors should consider revising this sentence.*

From a synoptic point of few, we would describe the atmosphere inside the ABL as turbulent and above as non-turbulent. But indeed the sentence as it was written is a simplification. We have clarified the wording.

*Page 9 - Figure 3 - I wonder if the source term modelling depicted in Figure 3 applies for both EXP1 and EXP2. This is not clearly mentioned in the paper. Can the authors clarify this point?*

For EXP1 and EXP2 individual source terms were obtained based on the corresponding measurements and used, whereby Fig. 3 only shows an example of the procedure.

*Page 9 - Line 19 - The number of numerical particles (100,000 in EXP1 and 270,000 in EXP2) used in the Itpas simulations seems to me quite low. I wonder if this number is enough at least in EXP2 with particles supposed to travel several hundreds of kilometers. Was a sensitivity study about the number of numerical particles carried out by the authors?*

There are no particle sinks in the model beside of the surface and the domain edges. Especially for the uplifting case, we do not have big losses once the particles start

to rise. We checked for the minimum necessary particle number and above 50k particles we do not see qualitative differences in the results. This means with at least 50 thousand particles we were able to produce similar horizontal trajectory patterns and vertical mixing like with 500 thousand or one million particles. Of course, a larger sample provides a more robust statistic but it also yields to higher computational costs. With a particle number with an order of magnitude of $10^5$, we found a good balance for the presented simulations.

*Page 10 - Line 5 - The source term in EXP1 and EXP2 is contained in a volume of about 10x10x5 m3. The initial distribution of the numerical particles in this volume is given by a detailed model. Is it really necessary to be so precise (see Figure 3) when considering the horizontal and vertical dimensions of the simulation domain for the atmospheric transport and dispersion?*

We need to define bubble-shaped volume calculate the expected number of mobilised particles from the measurement points. Inside the volume especially the vertical distribution is significant. The closer the particle starts to the surface the lower is there chance to get into the uplifting motion. The horizontal distribution inside the start volume is not as important as the vertical.

*Page 11 - Figure 4 - In EXP1, it is not clear for me if the particles are more or less lifted depending on their diameters. Can the authors clarify this point?*

Yes, in essence. During EXP1 the atmospheric conditions were such that the upward motion was low and lifting forces counteracting gravitational settling were minor. So the altitude the particles have shown in Fig. 4 is basically the height up to which the tractor's tool mechanically entrains the particles. In its model representation, the

height above ground particles can reach as a function of the particle size ultimately depends on the measured vertical profile for the different size classes. For your information, vertical profiles and concentration arrays for all size bins are included in the new supplement (Fig. S1 - Fig. S4)

*Page 11 - Figure 4 - Moreover, I'm concerned how the meteorological model can give information so close to the ground and along a so short horizontal distance (10 to 20 km) and height (5 m). As for me, there is an inconsistency between the space and time resolution of COSMO meso-scale weather forecast and the micro-scale transport and dispersion of the particles in EXP1. Explanations and justification from the authors are needed here! (I'm more confident with EXP2 simulation.)*

Thank you for bringing up this. We are agree that there is a conflict in the spatial resolution. In the transport range of this simulation there are other requirements for the model that COSMO cannot necessarily fulfill. This includes e.g. structures like buildings and forests that feedback on the wind. We have added a paragraph on the limitations. EXP1 should be considered as a case without transport. The particles were lifted mechanically and deposit again right after. Since it takes some time for the particles to settle down they get transported by the horizontal wind, but this transport is not reliable as described above.

*Page 12 - Figure 5 - I'm a bit surprised by the large horizontal expansion of the particles plume in Figure 5a even if explanations are given by the authors in the paper (development of the ABL and turbulent diffusion around noon - see Page 14 - Line 9). I'm even more surprised by the vertical ascending motion not of the smallest particles (less than 1 or 2 $\mu$m), but of the largest particles (up to 30 $\mu$m). Looking at Figure 5c, it seems that there are only very few differences between the aerodynamic behaviour of the smallest and largest solid particles. Can the authors comment on this point?*

The horizontal spread depends on numerous atmospheric conditions like discussed in the paper. In other test cases during the model development, we also saw more compact plumes. Also, vertical mixing is a decisive point for the horizontal dispersion. In situations with strong vertical mixing, like during our EXP2, vertical shear of wind speed and direction generally fosters horizontal dispersion.

In our understanding the even mixing of all particle sizes is plausible. As smaller particles have a smaller particle mass (assuming similar density), larger particles show higher settling velocities. The settling velocity for a particle with a diameter of $30\mu m$ is around 7.5 $cm\,s^{-1}$ and when the vertical wind velocity (mean wind + turbulent wind) has a higher value then the particle will rise (in the model's world). Vertical wind velocity in this order of magnitude is not unusual for a convective boundary layer like in our test case.

*Page 12 - Figure 6 - This is not easy at all to figure out the gradient of the virtual potential temperature just by looking at Figure 6. It would be simpler to determine the stratification of the ABL by visualizing vertical profiles of the temprature gradient graphed at successive hours of the day. I suggest the authors to add this information in supplementary materials.*

We agree and added a corresponding figure to the appendix (new Fig. A1)

*Page 14 - Line 4 - What is supposed to be evident is actually not so obvious. See my remark just before.*

It shall become evident that the boundary layer becomes unstable. However, we agree

that the sentence is ambiguous and we have clarified the wording.

*Page 14 - Line 19 - What are the computational times to simulate EXP1 and EXP2? Would it be possible to use Itpas off-line? What would be the difference in the computational times?*

Itpas is designed as an on-line application. In EXP1 where (nearly) all particle deposit within the first minutes the running time is comparable to a simulation without particles. EXP2 took roughly ten times longer than the simulation without particles. Our simulations of EXP2 on 36 processors (Intel(R) Xeon(R) Platinum 8160 CPU; 2.10GHz) took 260 minutes compared to 25 minutes without particles. But just for curiosity: If we could have the same data available as in the on-line coupling for an off-line run, we would expect that the simulation would take around 235 minutes on a serial process on the same server because of the insufficient parallelisation. Please note, that we added a new section about the model performance to the manuscript.

*Page 15 - Line 12 - Most of LPDM dedicated to local or regional scale simulations use the TKE to evaluate the variances of the velocity fluctuations and the Lagrangian time scales in the three spatial directions. This is probably less the case of LPDM adapted to larger scales like FLEXPART or HYSPLIT. The difficult point with TKE remains to distribute the turbulence between the space directions. Could be the authors recall how they proceed in Itpas (see my previous question about Page 6 - Line 3) and comment on this aspect?*

Please see above for an answer to this question.

*Page 15 - Line 16 - According to the authors, the area affected by agricultural emissions are "several times larger" than by wind erosion. This is not obvious. I*

*wonder why?*

Wind erosion can only take place in agriculture when sufficient wind and non-vegetated fields occur together on sandy soils. This can happen for a time period in spring and late summer/autumn. On the other hand, dust emission can also occur during agriculture-related activities, e.g. ploughing, spreading of fertiliser, harvesting or simply driving over the field with heavy machines. Then the emission does not depend on the wind. Depending on the atmospheric situation, this may mean that dust is only whirled up (EXP1) or that dust becomes airborne and gets transported away (EXP2).
* * *